# Look Who’s Talking: Host and Pathogen Drivers of *Staphylococcus epidermidis* Virulence in Neonatal Sepsis

**DOI:** 10.3390/ijms23020860

**Published:** 2022-01-13

**Authors:** Isabella A. Joubert, Michael Otto, Tobias Strunk, Andrew J. Currie

**Affiliations:** 1Centre for Molecular Medicine & Innovative Therapeutics, Murdoch University, Murdoch, WA 6150, Australia; isabella.joubert@murdoch.edu.au; 2Women’s and Infants Research Foundation Laboratory, King Edward Memorial Hospital, Subiaco, WA 6008, Australia; 3Wesfarmers Centre of Vaccines and Infectious Diseases, Telethon Kids Institute, Nedlands, WA 6009, Australia; tobiasstrunk@yahoo.de; 4Pathogen Molecular Genetics Section, Laboratory of Bacteriology, National Institute of Allergy and Infectious Diseases, U.S. National Institutes of Health, Bethesda, MD 20892, USA; motto@niaid.nih.gov; 5Neonatal Directorate, Child and Adolescent Health Service, Nedlands, WA 6009, Australia

**Keywords:** host–pathogen interactions, neonatal sepsis, *S. epidermidis*, commensalism, pathogenesis, virulence

## Abstract

Preterm infants are at increased risk for invasive neonatal bacterial infections. *S. epidermidis*, a ubiquitous skin commensal, is a major cause of late-onset neonatal sepsis, particularly in high-resource settings. The vulnerability of preterm infants to serious bacterial infections is commonly attributed to their distinct and developing immune system. While developmentally immature immune defences play a large role in facilitating bacterial invasion, this fails to explain why only a subset of infants develop infections with low-virulence organisms when exposed to similar risk factors in the neonatal ICU. Experimental research has explored potential virulence mechanisms contributing to the pathogenic shift of commensal *S. epidermidis* strains. Furthermore, comparative genomics studies have yielded insights into the emergence and spread of nosocomial *S. epidermidis* strains, and their genetic and functional characteristics implicated in invasive disease in neonates. These studies have highlighted the multifactorial nature of *S. epidermidis* traits relating to pathogenicity and commensalism. In this review, we discuss the known host and pathogen drivers of *S. epidermidis* virulence in neonatal sepsis and provide future perspectives to close the gap in our understanding of *S. epidermidis* as a cause of neonatal morbidity and mortality.

## 1. Introduction

Approximately 10% of infants are born preterm (defined as all live births <37 completed weeks of gestational age; GA) each year with numbers rising [1]. At the same time, advances in neonatal medicine have led to improved survival of preterm infants, including those born very and extremely preterm (<32 weeks GA and <28 weeks GA, respectively) [2]. However, prolonged hospitalization and invasive life support interventions are associated with an elevated risk of developing serious and life-threatening neonatal infections [3]. Bloodstream infections are the most common type of healthcare-associated infections, accounting for 11–19% of neonatal mortality worldwide [4]. Neonatal sepsis can be classified into early-onset sepsis (<72 h of life; EOS), which is usually caused by vertical transmission of maternal pathogens (i.e., Group B Streptococcus (GBS) and *Escherichia coli* (*E. coli*)) before or during birth [5], and late-onset sepsis (3–28 days of life; LOS) caused by nosocomial or community-acquired pathogens after birth [6]. LOS affects approximately in 1 of 10 very preterm infants (<32 weeks GA) with incidences reaching as high as 60% in the most extremely immature infants [7]. Gram-positive organisms are the most frequently isolated and in high-resource settings coagulase-negative staphylococci (CONS) are the predominant pathogens [8], accounting for up to 78% of LOS episodes [9]. Among CONS species, *Staphylococcus epidermidis* (*S. epidermidis*) is the most commonly isolated LOS pathogen in preterm infants [10]. Furthermore, *S. epidermidis* can initiate or aggravate other inflammation-associated morbidities in neonates, such as bronchopulmonary dysplasia, white matter injury, and retinopathy of prematurity [11]. In addition to maturity and birth weight, LOS risk factors include central venous catheters (CVC) intravenous lipids, mechanical ventilation, as well as prolonged use of antibiotics [6,12,13]. Early enteral feeding with unpasteurized breastmilk and supplementation with probiotics can reduce the risk of LOS in preterm infants [12,13]. While associated with lower mortality rates compared to sepsis caused by Gram-negative pathogens [14,15], short and long-term sequalae of *S. epidermidis* sepsis survivors include neurodevelopmental impairment and cerebral palsy [11]. The impact of *S. epidermidis* sepsis on long-term cognitive functions in preterm infants is still inconclusive [16,17]; however, persistent activation of neonatal inflammatory pathways, such as NF-κB, by *S. epidermidis* antigens may result in inflammatory disorders culminating in periventricular white matter injury [18].

### Diagnosis, Prevention, and Treatment of LOS

Rapid and accurate diagnosis of neonatal sepsis remains challenging. Infants with LOS usually display non-specific symptoms and clinical signs alone are notoriously unreliable [19]. The current gold standard for neonatal sepsis diagnosis are microbial blood cultures, which have a long turn-around time (12–48 h) [20]. Moreover, while blood cultures are specific, their sensitivity is highly dependent on the inoculated blood volume, which is limited (often less than 1 mL) in preterm infants and LOS often has low and intermittent levels of bacteraemia (7–121 colony-forming units (CFU)/mL) [20,21]. Among *S. epidermidis* LOS cases, one third presented with <50 CFU/mL [22]. As a ubiquitous skin commensal, *S. epidermidis* is also a common contaminant in culture-based diagnostics, which poses an additional challenge in the diagnosis of *S. epidermidis* sepsis [23].

Because of diagnostic deficiencies, broad-spectrum antibiotics are often administered to preterm infants upon first signs of systemic infections. Flucloxacillin, a penicillin, or vancomycin are most commonly administered [24]. Due to the emergence and global spread of nosocomial *S. epidermidis* strains with reduced sensitivity towards antibiotics [25,26], these infections may result in persistent bacteraemia in preterm infants, despite aggressive antibiotic treatment [27]. Among US NICUs, 78.6% of very preterm infants and 87% of extremely preterm infants receive antibiotics within the first 3 days of life [28]. The SCOUT study found that among NICU-admitted infants (34–39 weeks GA), only a small fraction (5%) treated with antibiotics had a culture-proven infection [29]. This suggests that unnecessary antibiotics use may occur in a large number of preterm neonates, which is associated with potential adverse outcomes for the infant, including impaired gut microbiome development/gut dysbiosis as well increased risk for subsequent bacterial infections, particularly necrotizing enterocolitis (NEC) [30,31].

The most effective prevention strategies of neonatal CONS sepsis include meticulous hand hygiene and early enteral feeding with unpasteurized breast milk [32]. Furthermore, neonatal skin care with topical emollients, such as coconut oil, maintains skin condition in very preterm infants during the first weeks of life [33]. Except for the use of probiotics, no adjunct therapies for the treatment and prevention of neonatal infections have shown direct benefit [34].

Understanding the unique host–pathogen interactions between preterm infants and nosocomial CONS infections, particularly the dominant organism, *S. epidermidis,* is essential for the development of alternative and targeted preventative, diagnostic, as well as therapeutic tools to combat neonatal infections with this opportunistic pathogen.

## 2. *S. epidermidis*: Friend and Foe

### 2.1. S. epidermidis as an Early and Ubiquitous Colonizer of the Newborn

*S. epidermidis* is a commensal bacterium and can be found ubiquitously on human skin and mucous membranes of the respiratory tract and intestine [35,36,37]. The human fetus may encounter *S. epidermidis* already in the uterus, as recently proposed by Stinson et al., who found *Staphylococcus* spp. present in amniotic fluid using high resolution full-length 16S rRNA gene sequencing [38]. The initial colonization of preterm infant skin and intestine with *S. epidermidis* likely commences immediately after birth via contact with parents, hospital staff and equipment [39,40]. In the case of preterm infants, premature rupture of membranes (PROM) and intra-amniotic infection (chorioamnionitis) furthermore lead to microbial exposure before birth in 25–30% of infants [41]. The development of the early life neonatal skin and gut microbiome is GA-dependent and partly shaped by mode of delivery, feeding, and administration of antibiotics [42]. *S. epidermidis* is abundant in the colostrum and breast milk of healthy women [43,44]. Breast milk of preterm infant mothers contains higher counts of *S. epidermidis* which harbour virulence-associated genes, which may be due to prolonged hospital exposure [45]. *S. epidermidis* colonization of the preterm infant gut is promoted by enteral feeding with unpasteurized mothers’ milk and limits the intestinal spread of NICU-associated *S. epidermidis* strains [46]. Notably, feeding with bovine colostrum immediately after birth improved the clearance and clinical responses to *S. epidermidis* sepsis in a preterm piglet model [47].

The microbiome of preterm infants, which spend a prolonged time in the hospital, has been suggested to be additionally shaped by the “NICU microbiome”, consisting of a low number of mostly skin-associated bacterial taxa [48]. As such, preterm infants generally exhibit lower abundances of commensal, obligate anaerobic bacteria, but a higher prevalence of nosocomial isolates and multi-drug resistant strains, with enrichment of *S. epidermidis*, *Enterococcus faecalis*, and *Klebsiella pneumoniae* compared to term infants [49,50]. Colonization with *S. epidermidis* has been shown to be established by multiple founder lineages which form specialized communities at different skin sites (i.e., moist, oily, and dry) in a dynamic manner with shifts in intra-species abundances during the first month of life [51,52]. This diversity is maintained under high selective pressure throughout life with *S. epidermidis* strains continuously being transmitted between different skin sites [52].

### 2.2. Establishing Host Tolerance to Commensal S. epidermidis

The concept of disease tolerance as a defence strategy against infectious agents in animals was championed by Medzhitov et al. as an alternative to disease avoidance and resistance, which are associated with increased energetic and immune cost [53]. More recently, the concept was revisited by Harbeson et al. through the perspective of neonatal host defences in early life [54]. In brief, the authors suggest that the first postnatal weeks of life constitute a state of heightened immunogenic susceptibility to bacterial pathogens where the neonatal host must carefully balance between energy expended for immune defences (i.e., disease resistance) and the energy required for growth and development (i.e., resulting in increased disease tolerance). This is supported, for instance, by the observation that neonates exhibit a higher bacterial burden (CFU/g lung) and a delayed clearance of bacteria compared to adults in a murine model of methicillin-resistant *Staphylococcus aureus (S. aureus)* (MRSA)-pneumonia [55].

Additionally, the neonatal period is a time of active adaptive immunosuppression, with high abundances of regulatory T cells (Tregs) present in neonatal skin as well as peripheral tissue (Figure 1) [56]. A specialized subset of Tregs induces tolerance to *S. epidermidis* colonization during neonatal skin development, but not during adulthood [57]. Moreover, colonization with *S. epidermidis*, but not *S. aureus,* heightens Treg responses in neonatal mice [58]. Of note, Tregs in preterm infants have a distinct phenotype with heightened immunosuppressive capacity compared to term infants [56], and are implicated in the development of bronchopulmonary dysplasia and may play a role in EOS [56,59] by temporarily down-regulating host immune functions and facilitating increased bacterial establishment. Other subsets of neonatal cells contribute to inhibit hyperinflammation in response to rapid commensal colonization of the skin and mucosal membranes after birth, including CD71+ cells (immature erythrocytes) and intestinal epithelial cells (IECs) [60,61]. In their immunosuppressive role, CD71+ cells are also implicated in compromising innate and adaptive immunity against bacterial infections, for instance, with *Listeria monocytogenes* and *Bordetella pertussis* [62,63]. However, in a murine model of neonatal polymicrobial sepsis, ablation of CD71+ cells did not alter sepsis mortality [64]. Immunosuppressive molecules, such as S100-type alarmins (i.e., S100A8 and S100A9) also play a role in the postnatal gut microbial development by preventing excessive inflammation, with S100-knockout mice showing higher neonatal sepsis mortality [65,66,67]. Notably, preterm infants show reduced S100A8/S100A9 levels compared to term infants [67].

In conclusion, transient neonatal immunosuppression is crucial to establish tolerance against neonatal commensals, but simultaneously contributes to the elevated neonatal susceptibility to infectious disease in early life. However, the specific role of immunosuppressive cell subsets and molecules in regard to early life *S. epidermidis* colonization remains to be fully elucidated.

### 2.3. Contribution of S. epidermidis to Host Defence

Apart from direct competition for nutrients and host adhesion sites, commensal bacteria produce specific metabolites and antimicrobial peptides to inhibit pathogenic colonization [68]. In its commensal role, *S. epidermidis* modulates the host immune system and shapes the development of the skin and nasal microbiome [37,69] by preventing host colonization by more virulent bacterial and fungal organisms, as well as viruses (Figure 1) [70,71,72]. *S. aureus* nasal colonization, for instance, is inhibited by commensal *S. epidermidis* through the production of biofilm-disrupting compounds, such as serine protease Esp, which degrade *S. aureus*-specific host receptor proteins [73,74]. In hospitalized neonates, nasal *S. epidermidis* colonization has been shown to protect against *S. aureus* colonization by a Esp-independent mechanism [75].

Toll-like receptor (TLR) 2 stimulation by *S. epidermidis* induces the production of host antimicrobial peptides by keratinocytes and nasal epithelial cells to boost host immunity [39,74,76]. In a recent study, Pastar et al. reported the elevated ability of skin-resident γδ T cells in killing intracellular *S. aureus* following stimulation with *S. epidermidis*, via cell-specific upregulating of host perforin-2 (P-2) production [69]. Moreover, *S. epidermidis* induces epidermal IL-17-producing CD8+ T cells (Tc17) to upregulate S100-type alarmins, which limits pathogen invasion and promotes wound healing [70,77].

By potentially reducing the survival of opportunistic pathogens such as *S. aureus* and *Streptococcus pyogenes* through the secretion of antimicrobially active phenol-soluble modulins (PSMs) [76,78], *S. epidermidis* may protect from disease, such as atopic dermatitis (AD) [79]. However, while in vitro studies using high concentrations of synthetic PSM peptides reported antimicrobial activities, in another study, only proteolytically cleaved but not intact peptides exhibited activity against *S. pyogenes* [80], with their in vivo role and biological significance still unclear [81]. Furthermore, a more recent study by Cau et al. found that *S. epidermidis* colonization was associated with more severe AD disease [82], which may be due to S100A8/A9-dependent alterations of the skin barrier proteins [83].

Antiviral activities of *S. epidermidis* have also been documented. For instance, nasal commensal *S. epidermidis* can act as a frontline protection against influenza A virus through the upregulation of IFN-λ production in the nasal epithelium [71].

Lastly, a commensal *S. epidermidis* strain has recently been suggested to protect from UV-induced skin neoplasia by inhibiting DNA polymerase activity selectively in tumour cells through producing a nucleobase analogue [84].

### 2.4. S. epidermidis as a Human Pathogen

Compared to *S. aureus*, an important community pathogen causing skin and soft tissue infections [85], *S. epidermidis* is considered a low virulence organism and is typically non-haemolytic in human blood. It also does not express potent toxins or exoenzymes known to be determinants of *S. aureus* pathogenicity [86,87].

PSMsare a family of amphipathic peptides produced by *S. epidermidis* with multifactorial roles in its commensal as well as pathogenic niches [81]. PSMs differ in their length and net charge, with Staphylococci species each having a specific PSM profile [88]. Known PSMs produced by *S. epidermidis* include PSMα, PSMδ, PSMε, δ-toxin (PSMγ), PSM-*mec*, and two PSMβ peptides (PSMβ1, PSMβ2), which have been attributed a range of roles, including biofilm formation and dispersion, evasion of host immunity, and interspecies competition [81,89]. While some PSMs are highly cytolytic (i.e., α-type PSMs and PSMδ), in *S. epidermidis*, these are expressed in lower amounts and with reduced activity [81].

In accordance with its role as a ubiquitous commensal, *S. epidermidis* rarely causes invasive disease in healthy adults, children, or even term-born infants. However, opportunistic healthcare-associated infections with *S. epidermidis* are on the rise among adult patients with indwelling devices [90]. Biomedical device-related *S. epidermidis* infections are often of chronic nature and difficult to treat with antibiotics due to *S. epidermidis* strong abilities to produce biofilm on biotic and abiotic surfaces [91,92,93]. As a persistent inhabitant of skin and mucosal tissue, *S. epidermidis* has evolved diverse mechanisms for host attachment and accumulation, which contribute to both commensalism and pathogenicity [86]. This includes host protein-specific surface adhesion molecules, non-specific adhesion mechanisms (i.e., hydrophobic interactions), cell-wall-anchored proteins/adhesins (i.e., microbial surface components recognizing adhesive matrix molecules; MSCRAMMs), as well as peptidoglycan-bound wall teichoic acid (WTA) [94]. The production of extracellular polymeric substances (EPS) is considered a key virulence factor of *S. epidermidis* and has been shown to facilitate its attachment to medical devices during hospital-associated infections [91,93].

The ability to produce a mature biofilm is facilitated by complex multi-step mechanisms and is dependent on the presence of selected genes, particularly the intercellular adhesion (*icaADBC*) operon and insertion element IS*256* [95,96], which are involved in the synthesis of the main biofilm exopolysaccharide polysaccharide intercellular adhesin (PIA) [97], also known as poly-N-acetylglucosmine (PNAG), as well as accumulation-associated proteins, such as extracellular matrix binding protein (Embp), biofilm associated-homologous protein(Bhp), and accumulation-associated protein (Aap) [98,99,100]. *S. epidermidis* secreted peptidoglycan hydrolases (i.e., autolysin AltE and autolysin/adhesin Aae), which play a role in the cellular processes involved in bacterial proliferation (e.g., cell separation), also promote primary attachment to medical devices [101,102]. Furthermore, alternative morphotypes, assembled from extracellular DNA (eDNA, released from dying cells) and (lipo)proteins have been described [103,104].

During changing environmental conditions, *S. epidermidis* uses the accessory gene regulator (*agr*) quorum-sensing system to detect increases in population density [105,106]. The *agr* system is encoded by the *agrBDCA* operon and multiple different *agr* types have been described in *S. epidermidis* [107]*. S. epidermidis* quorum-sensing facilitates the cross-inhibition of *S. aureus* [108]. Moreover, the colonization of a host niche by multiple *S. epidermidis agr* types has recently been suggested to suppress the expression of virulence factors to maintain homeostasis [52].

*S. epidermidis* PSMs, which are under strict control of the *agr* operon [109], play putative roles in biofilm maturation as well as in vivo biofilm dissemination [110,111]. Notably, *S. epidermidis* biofilm formation in response to harsh environmental condition, is accompanied by an extensive change in gene expression which facilitates the change from planktonic growth to quiescent mode, characterized by low expression of virulence factors, such as PSMs and *agr* [112].

A variety of mechanisms are employed by *S. epidermidis* to evade physical and chemical threats elicited by host innate and adaptive immunity [113,114]. A versatile range of antimicrobial peptides (AMPs) are produced by the host as a first line of antibacterial defence on human skin [115]. While lower in total host defence proteins, neonatal skin harbours higher concentrations of AMPs compared to adult skin [116,117]. *S. epidermidis* biofilm formation is a key mechanism to evade host immunity [118], including host AMPs [119], complement compound C3b and immunoglobulin G (IgG) [120]. *S. epidermidis* has also developed AMP sensor systems (*aps*) [121], efflux pumps for specific bacteriocins (*vraFG* ABC transporters) [122], and extracellular proteases (i.e., metalloprotease SepA) [114] to provide protection from host AMPs, complement components, and avoid neutrophil killing.

## 3. *S. epidermidis*: Inside, Outside—Everywhere?

As a human commensal species, *S. epidermidis* is ubiquitously present in the community as well as in the hospital environment, where healthcare workers serve as an important nosocomial reservoir for *S. epidermidis*, including methicillin-resistant *S. epidermidis* (MRSE) strains [123]. While invasive procedures are assumed to be a common route of entry for *S. epidermidis* in preterm infants, not all cases of *S. epidermidis* LOS are device-related and specific bacterial DNA signatures are often undetectable from removed material [124].

Emerging evidence suggests that the gut is an important alternative reservoir of LOS-causing *S. epidermidis* in preterm infants with immature intestinal mucosal barriers and bacterial dysbiosis (Figure 2) [125]. Disturbances in the development of mouth, gut, and lung microbiomes of preterm infants can be caused by common empiric antibiotic exposure, lack of enteral feeding, as well as use of proton pump inhibitors [126,127]. A disturbed intestinal microbiome can impair the development of colonic Treg cells, resulting in dysregulated T cell responses upon bacterial challenge [128]. Innate immune responses and IL-17 producing type 3 innate lymphoid cells (ILC3) in particular, play a critical role in immune homeostasis of the intestinal tissue during the perinatal period [129]. Even short-term antibiotic exposure can decrease ILC3s and IL-17 levels in neonatal mice resulting in increased bacterial translocation from the gut lumen and heightened susceptibility to LOS [130]. A reduced abundance of ILC3s may be due to decreased TLR signalling by gut commensals. In septic preterm infants the gut microbiome is less diverse compared to matched controls. Particularly increased carriage of *Staphylococcus* spp. and Enterobacteria and lower numbers of Bifidobacteria [131], may impair epithelial barrier function [132,133].

Direct evidence of intestinal *S. epidermidis* causing LOS is scarce; however, the presence of *S. epidermidis* has been detected in ~40% of preterm infant meconium samples and ~90% of faecal samples (collected 7 days after birth) [134]. Moreover, *Staphylococcus* was found more abundant in preterm infant faecal samples before and during sepsis [131]. However, a study by Stewart et al. found that while the dominant taxa in the gut microbiome of preterm infants with LOS was usually identical to the causative LOS pathogen identified through blood culture, when sepsis with *S. epidermidis* was diagnosed, *Klebsiella* and *Escherichia* dominated the intestine [135]. Most recently, Golinska et al. have found that 25% of CONS strains isolated from the bloodstream of VLBW/LWB neonates were identical to the strains isolated from faeces of the same neonates at the same time [125].

## 4. Genetic Determinants of Virulence in Commensal and Invasive *S. epidermidis*

*S. epidermidis* comprises a large number of clonal lineages with a pan-genome of about 2.5 Mb, consisting of approximately 80% core genome (containing genes present in all strains) and 20% accessory genome (genes present in only a subset of strains) [136,137]. The *S. epidermidis* pan-genome is open, indicating that the species undergoes frequent horizontal gene transfer (HGT) [138]. Bacteriophages are the major vehicles for HGT in staphylococci, governing the evolution of new clonal lineages [139,140]. Unlike other bacterial species, *S. epidermidis* isolates are genetically diverse with a large collection of genetic determinants for resistances against antibiotics and host immune defence (i.e., AMPs) [137]. *S. epidermidis* shares approximately half of its genome with its more virulent cousin *S. aureus*, and interspecies HGT of mobile genomic elements has been well documented between the species, including genes encoding for metal detoxification and enhanced pathogenicity [141]. *S. epidermidis* is an important reservoir of drug resistance genes, which they can transfer to *S. aureus* and *Enterococcus* spp., thereby driving the emergence of more virulent hospital-associated strains, particularly methicillin-resistant *S. aureus* (MRSA) [142,143]. The high genetic heterogeneity and rapid adaption through intra- and interspecies HGT underlies *S. epidermidis* ability to colonize and persist in various host niches and host populations [38,142,144].

### 4.1. Nosocomial S. epidermidis Strains Implicated in Neonatal Infection

*S. epidermidis* isolates can be grouped into two main phylogenetic genotype clusters (clonal complexes), lineage A/C (comprising most nosocomial isolates) and lineage B (mostly community isolates) [136,141]. Both clusters display a high genome plasticity with specific functions, suggested by the presence of a high number of genes exclusive to each cluster [145].

Multi-locus sequence typing (MLST) allows tracking of the evolution of *S. epidermidis* populations, and rise of new variants through mutations and HGT [146]. In hospital environments, two sequence types (STs), ST2 and ST23, account for most adult clinical disease (i.e., bacteremia, CVC infection, septic arthritis) [147]. Furthermore, having an ST2 or ST5 *S. epidermidis* bacteraemia has been found to be an independent predictor of complicated bloodstream infection [148]. Neonates in the NICU are most frequently colonized with *S. epidermidis* strains belonging to ST2, ST5, ST59, and ST81, and among LOS-causing isolates, ST2 and ST5 are most commonly identified [144].

Since the advent of next-generation sequencing, a vast collection of clinical and commensal *S. epidermidis* isolates have been sequenced [54,142,149]. By analysing the genetic determinants of isolates associated with asymptomatic carriage and disease, numerous studies have aimed to discover discriminating markers of *S. epidermidis* invasiveness among adults and neonates [138,149]. While there is some evidence for *S. epidermidis* adaptations within the host [150] and some differentially distributed virulence factors may contribute to invasiveness, previous studies were not able to conclusively pinpoint a genetic factor solely associated with virulence vs. commensalism in *S. epidermidis* [136,137,138,145,151]. Of note, a recent study by Du et al. has described a mutation in the *tarIJLM* gene cluster, which inhibited *S. epidermidis* capacity to colonize epithelial cells but promoted binding to endothelial cells, through changes in wall-teichoic acid structures expressed [140]. This mutation facilitated bloodstream invasion and increased sepsis mortality in a mouse model and was found only among a small number of infection-associated *S. epidermidis* isolates (6.9–13.9%), with increased prevalence in ST2, ST5, ST10, ST23, and ST87 strains.

In the next paragraphs, we will discuss some recent comparative genomics studies in adult and, where available, neonatal clinical and commensal *S. epidermidis* isolates, and put their findings into the context of previous key functional studies of *S. epidermidis* virulence.

### 4.2. The Role of S. epidermidis Biofilm in Neonatal Invasive Disease

*S. epidermidis* virulence in hospital-acquired infections is thought to be driven mainly by their ability to colonize the surfaces of medical devices and form persisting biofilm, thereby evading the host immune responses and antibiotic treatment [118]. Indeed, biofilm-associated genes, such as the *icaADBC* operon and IS*256* are more prevalent in distinct clonal complexes and ST types (i.e., ST27), comprising mostly clinical *S. epidermidis* strains [152], and have been suggested as potential markers of invasive *S. epidermidis* [153,154]. Notably, *ica*-positive *S. epidermidis* strains are less adapted to human skin colonization [155] and the production of PIA and AltE is associated with an increased potential to cause bacteraemia in a rat model of central venous catheter (CVC)-associated infection [101]. In neonatal cord blood, *S. epidermidis* strains positive for *icaADBC* inhibit the release of IL-6, which was not observed in *icaADBC*-negative strains [156].

While the *agr* quorum-sensing system of *S. epidermidis* controls several putative determinants of acute virulence such as PSMs, proteases, and lipases [86,157], it reduces *S. epidermidis* primary host attachment and biofilm forming capabilities, but increases the direct virulence through the production of delta-toxins [158]. In line with this, *agr* non-functionality is more common among *S. epidermidis* strains isolated from joint protheses infections and *S. epidermidis agr* deletion mutants have increased ability to cause in vivo infection in a rabbit model of indwelling medical device-related infection [159]. By regulating the expression of pro-inflammatory PSMs, *agr*-positive *S. epidermidis* elicit elevated host immune responses and have been suggested to have increased invasive capacity [94,113,156,160].

Among *S. epidermidis* isolates from neonatal blood, the vast majority (85–90%) are capable of producing biofilm [161,162]. However, the rate of biofilm production and the presence of *icaADB* and *altE* in *S. epidermidis* isolates from neonatal bloodstream infection (BSI) was comparable to commensal *S. epidermidis* isolates from neonatal nares [149]. Furthermore, the rates of biofilm formation were comparable between persistent and nonpersistent neonatal *S. epidermidis* isolates upon growth in low-glucose media but higher in persistent isolates when grown in total parenteral nutrition (9.23% glucose) [163]. Similarly, *icaA* and other biofilm-associated genes, including *atlE, embp, mecA*, and IS*256*, were found equally prevalent in isolates of adult catheter-related BSI isolates compared to commensal isolates from healthy individuals [164], suggesting that biofilm-forming capabilities are not a distinguishing feature of *S. epidermidis* invasiveness among clinical and commensal isolates.

Investigating the role of biofilm formation during in vivo infection remains challenging, especially in the neonatal context, with only few established mouse and rabbit models. Moreover, the functional redundancy of *S. epidermidis* biofilm mechanisms complicate such studies [94]. The contribution of biofilm formation in acute neonatal sepsis and the neonatal immune response to *S. epidermidis* biofilm remains to be fully elucidated. However, biofilm-released *S. epidermidis*, such as from biomedical devices, have been established as an important source of bacteria seeding the bloodstream and causing acute sepsis [110,160]. Furthermore, the attenuation of host defences, particularly bacterial phagocytosis, by biofilm-producing *S. epidermidis* has been reported [122,165,166] and may inhibit the efficient clearing of bacteraemia in neonates.

### 4.3. The Role of S. epidermidis Host Adhesion in Neonatal Invasive Disease

A range of other putative *S. epidermidis* virulence determinants are likely involved in the onset and progression of LOS in the preterm infant (Figure 3). Indeed, the presence of genes encoding for specific surface adhesions have been suggested as discriminative markers for *S. epidermidis* invasiveness and virulence among neonates and adults, including *Staphylococcus epidermidis* surface protein I (*sesI*) [149,167]. Interestingly, *sesI* is mostly absent in *S. epidermidis* from healthy individuals [167] and is exclusively present in strains belonging to disease-associated *S. epidermidis*, such as LOS-causing ST2 clones [168]. *sesI* is also more prevalent in *S. epidermidis* strains harbouring other virulence-related genes, such as biofilm-associated genes *aap* and IS*256* in adults and neonates [149,168]. Using in vitro adhesion and aggregation assays, *sesI* was found to promote the initial stages of bacterial agglomeration and biofilm-formation [168]. However, the correlation of *sesI* with bacterial pathogenicity remains unclear and further studies in the neonatal setting are needed to understand their contribution to *S. epidermidis* virulence in vivo [169,170].

The Ser-Asp-rich fibrinogen binding protein-encoding genes *sdrF*, *sdrG (Fbe),* and *sdrH* contribute to *S. epidermidis* adhesion to extracellular host matrix molecules [171] and *sdrF* has been found enriched among neonatal bloodstream isolates (65%) compared to nasal isolates (11%) [149]. Notably, SdrF plays a key role in initiating driveline-associated *S. epidermidis* infections in a murine model and may facilitate transcutaneous entry of commensal *S. epidermidis* [165]. In a rat model of intravascular-catheter-associated infection, *sdrG*-positive *S. epidermidis* were more likely to cause bacteraemia and metastatic disease than *sdrG*-deficient mutants [166].

The role of other adhesion-associated surface proteins and enzymes Embp, Aap, Bhp, small basic protein (Sbp), and glycerol ester hydrolase (Geh) in facilitating *S. epidermidis* attachment to biotic and abiotic surfaces has been investigated in vitro [172]; however, their role in vivo is less well defined [173]. In conclusion, while some adhesion factors seem to be associated with bloodstream infections, their contribution to neonatal sepsis pathogenesis remains incompletely understood.

### 4.4. The Role of S. epidermidis Toxins in Neonatal Invasive Disease

While *S. epidermidis* biofilm production is considered a key virulence factor contributing to the prevalence of biomedical device-associated infections, *S. epidermidis* sepsis is acute in nature and other virulence factors as well as host immunity (or lack thereof) likely drive infections in the neonatal host (Figure 3) [11].

*S. epidermidis* is a low virulence organism with limited toxin production apart from PSMs [81]. Qin et al. have linked MRSE PSM-*mec* to bacterial survival in human whole blood and resistance to neutrophil killing [174]. Moreover, the secretion of PSM-*mec* toxins increased cytokine expression (IL-1*β*, TNF-*α* and the mouse IL-8 homologue CXCL1), leading to elevated sepsis severity in a murine model of sepsis [174]. The secretion of another PSM, haemolytic *δ*-toxin (PSM*γ*), by *S. epidermidis*, has been associated with the development of neonatal NEC [175]. However, there is no evidence yet for any specific virulence factor or toxin contributing to *S. epidermidis* sepsis pathogenesis in neonatal animal models using isogenic deletion mutants.

### 4.5. S. epidermidis Antibiotic Resistance and Neonatal Invasive Disease

Nosocomial *S. epidermidis* strains have adapted under selective pressures in the hospital environment, resulting in multiple resistance determinants and resistance-conferring mutations. A large majority of *S. epidermidis* isolates from preterm neonatal infections are MRSE, carrying the *mecA* gene on the staphylococcal cassette chromosome *mec* (SCC*mec*) [176]. Furthermore, resistance against other aminoglycosides (e.g., gentamicin), and macrolides (e.g., erythromycin and clindamycin) was reported recently among *S. epidermidis* isolates from NICUs across Europe [138,168,177]. Linezolid and rifampicin resistance can be conferred through mutations in ribosomal gene *rpoB* (RNA polymerase) and the presence of plasmid-derived *cfr* (RNA methyl transferase*),* whose prevalence are rising among *S. epidermidis* ICU strains [27,28,151]. A potential relationship between rifampicin resistance and vancomyin/teicoplanin resistance in *S. epidermidis* strains with dual *rpoB* mutations (D471E and I527M; most commonly found in ST2 and ST23 lineages), has been suggested [147]. This may be driven, in part, by device impregnation with antibiotics such as rifampicin, as lower rates of resistance has been observed in countries where this is not common practice [147,178]. However, *S. epidermidis* isolates from neonates remain largely susceptible to vancomycin and linezolid [168,177,179].

Nosocomial *S. epidermidis* strains are more likely to cause invasive disease in both adults and neonates, and can disseminate between hospitals and even between countries [180,181]. In a recent study using comparative genomics, *S. epidermidis* isolates from adult prosthetic joint infections more frequently harboured resistance against antibiotics, including β-lactams and aminoglycosides, and chlorhexidine, compared to commensal nasal isolates from patients with planned surgery [178]. However, Morgenstern et al. found no difference in the treatment cure rate of device-related infection caused by methicillin/multi-drug resistant vs. susceptible *S. epidermidis* [182]. Moreover, in neonates, the presence of antibiotic resistance had no effect on neonatal C-reactive protein (CRP) levels [183], suggesting that antibiotic resistance does not influence the clinical course of neonatal LOS. Currently, the impact of different antibiotic resistances in *S. epidermidis* on neonatal sepsis onset, disease progression and outcome require further research and clarification.

## 5. Phenotypic Characteristics of *S. epidermidis* Clinical Isolates

Upon host invasion, *S. epidermidis* undergoes a rapid change of environment, including alterations in nutrient availability, variations in temperature, salt and pH, the presence of circulating host immune cells, pro-inflammatory molecules, and antibiotics. Invasive *S. epidermidis* isolates have demonstrated ability to transition from growth on skin to growth in blood or in the presence of blood components [184]. In line with their elevated pathogenic potential, *S. epidermidis* strains of the A/C lineage display increased ability to resist oxidative stress, evade host immunity and resist antibiotics in conditions mimicking blood infection, compared to less virulent B lineage strains [145]. Furthermore, growth rate in the presence of human plasma has been reported as a key predictor of whether an isolated *S. epidermidis* strain originated from true bacteriemia vs. contamination [184,185].

Méric et al. found that disease-associated *S. epidermidis* isolates induce higher levels of IL-8 production by blood cells (but not keratinocytes), compared to carriage-associated isolates [138]. *S. epidermidis* also induces higher IL-8 production by small airway epithelial cells in vitro compared to the more virulent *S. aureus* [177]. This may contribute to the development of bronchopulmonary dysplasia by promoting the persistent migration of inflammatory cells into the neonatal lungs [177].

A trend towards reduced toxicity with disease-associated *S. epidermidis* isolates was observed in vitro, as determined by an vesicle lysis test specific for staphylococcal PSM toxins [138]. This is in line with previous findings, which report a correlation between reduced bacterial toxicity and more severe disease in studies of MRSA bacteraemia and pneumonia [179,186], which has been suggested to be a trade-off for increased fitness in human serum in these strains [187]. Underlying the observed association between lower toxicity and increased invasiveness among clinical *S. epidermidis,* may be the predominance of *agr*-negative *S. epidermidis* strains among invasive isolates [159], which produce less PSMs and have increased biofilm-forming capacities [105,158] and hence, are more frequently implicated in biomedical device-associated infections.

## 6. *S. epidermidis* Gene Expression during Neonatal Host Invasion

Genotyping and phenotypically characterizing clinical *S. epidermidis* isolates is key in tracking the emergence of novel strains and lineages associated with nosocomial infections, as well as their antibiotic resistance, to guide appropriate treatment. Unfortunately, this is usually only done for genetic studies of nosocomial pathogens rather than as a part of clinical routine. Moreover, these analyses do not consider bacterial gene expression, posttranslational modifications, or novel gene mutations with unknown impact on the virulence phenotype. Ultimately, the virulence factors actively driving infection are likely dependent on the unique in vivo interplay between host and bacterium, which include not only direct bacterial virulence, but the host immunological and metabolic environment as well as the local host microbiota [188,189,190].

Invasion of host tissue constitutes a substantial change of environment for (commensal) pathogens. As such, bacterial exposure to blood and blood components is accompanied by a major ‘metabolic reset’, with the upregulation of bacterial genes for amino acid biosynthesis, transport, and metabolism [191]. Human plasma has been suggested as the main driver of *S. epidermidis* transcriptional responses as well as biofilm formation in a human blood challenge model [192]. Furthermore, the availability of exogeneous essential nutrients is crucial for *S. epidermidis* growth and biofilm formation [193,194]. The host strongly limits availability of nutrients, such as iron, that are essential for bacterial growth, a phenomenon that has been termed “host nutritional immunity” [195]. Hence, it is not surprising that genes encoding for iron sequestration have been found to be rapidly upregulated by *S. epidermidis* biofilm upon blood exposure [196]. Genes encoding for biotin metabolism, an essential cofactor in central pathways for bacterial survival, are also significantly upregulated in *S. epidermidis* in response to blood and plasma and may comprise a therapeutic target to inhibit *S. epidermidis* growth [192].

The only study, to our knowledge, investigating the transcriptional responses of *S. epidermidis* from infection isolates was conducted on adult patients with postoperative endophthalmitis [197]. This work identified iron and pyruvate metabolic genes and staphylococcal toxin SE1634 as potential contributors of *S. epidermidis* pathogenesis during endophthalmitis [197]. However, RNA-sequencing in this study was not performed on the bacteria directly isolated from the infection site, but after growth on blood agar plates, which does not completely reflect in vivo conditions.

Studies analysing the whole *S. epidermidis* transcriptome upon interaction with human blood remain scarce, focusing primarily on *S. epidermidis* biofilm virulence [198,199,200,201]. Furthermore, there are no studies to date using neonatal *S. epidermidis* challenge models to understand bacterial transcriptional responses to blood in this vulnerable host population.

## 7. The Neonatal Host as an Important Contributor to *S. epidermidis* Virulence

Neonatal blood cell immune responses elicited by clinical and reference *S. epidermidis* strains (live or heat-inactivated) have been investigated in a collection of in vitro (term and preterm neonatal blood samples) and in vivo (murine) studies, as previously reviewed by Dong et al. [11]. Briefly, the neonatal immune response against *S. epidermidis* is mediated by innate immune responses, characterized by high levels of pro-inflammatory cytokines (i.e., IL-6, IL-8, TNF-*α*), which are GA-dependent, and lack regulatory control. As recently reported by Hibbert et al., sepsis risk in neonates may be driven by a stark initial hyperinflammatory immune response combined with simultaneous immunosuppression caused, amongst other factors, by Tregs, MDSCs, and T cell inhibition [202]. While some aspects of preterm immunity may have a reduced functional capacity upon pathogenic stimulation (i.e., AMPs, reactive oxygen species, complement elements) compared to term infants and adults, *S. epidermidis* phagocytosis by preterm monocytes was not observed to be dysfunctional [11,203].

Of note, cytokine responses to *S. epidermidis* have recently been shown to be impaired in the whole blood of preterm infants who develop LOS, before clinical onset of their disease [198], suggesting that functional differences in immune responses play a role in the development of preterm *S. epidermidis* LOS.

*S. epidermidis* sepsis isolates may induce a greater oxidative burst compared to non-clinical reference strains; however, Dong and Speer et al. have stressed that the strain characteristics available are often unspecific [11]. When compared to in vitro challenge with other sepsis-causing pathogens such as *E. coli* or GBS, *S. epidermidis* induces similar or lower pro-inflammatory responses [199].

We are aware of four studies, to date, investigating the (preterm) neonatal host immune response and neurodevelopmental outcomes using a mouse model of *S. epidermidis* sepsis [200,204,205,206]. All studies demonstrated inoculum dose-dependent (TLR2-dependent and independent) inflammation with markedly increased pro-inflammatory cytokines and chemokines in blood and central nervous system (CNS; i.e., brain or cerebral spinal fluid; CSF), resulting in white and grey matter injury post challenge in one study. This heightened inflammatory state may explain the CNS injury associated with *S. epidermidis* infections in preterm infant [207]. Interestingly, *S. epidermidis* infection resulted in increased susceptibility to hypoxia-induced ischemia only in male but not female mice [200]. All studies used *S. epidermidis* 1457, an invasive clinical strain obtained from an adult catheter infection [201].

On a transcriptional level, LOS (regardless of causative pathogen) has been shown to induce a significant shift in gene expression, with elevated expression of IFN-α/β, IFN-γ, IL-1, and IL-6 pathways among very preterm infants [208]. Notably, among VLBW preterm infants, different gene expression clusters (‘endotypes’) corresponding to those detected in Gram-positive sepsis and Gram-negative sepsis were observed [209]. Cord blood monocytes from very preterm and term infants challenged with live *S. epidermidis* and *E. coli* shared a transcriptional response, driven mainly by TLR/NF-κB/TREM-1 signalling [210]. However, *E. coli*-stimulated, but not *S. epidermidis*-stimulated neonatal monocytes showed a GA-independent upregulation of IFN genes [210].

Collectively, these studies suggest that in addition to preterm infants exhibiting distinct immune capacities compared to term infant and adults [211,212], *S. epidermidis* elicits a potent and specific immune response in preterm neonates, compared to term infants. That alone, however, does not explain the unique susceptibility of preterm infants to *S. epidermidis* infections and their long-term adverse outcomes, as only a subset of the preterm infant population will develop serious infection in response to exposure to NICU *S. epidermidis* strains. Hence, the response of *S. epidermidis* to exposure and invasion of the preterm neonatal host may play a significant part in this interplay and understanding the bacterial drivers of preterm neonatal invasion in vivo will likely shed new insights into this tug-of-war.

## 8. Conclusions and Future Perspective

The adaptation of *S. epidermidis* is highly dynamic and directly shaped by the specific selection pressures encountered in the host environment. Notably, within-host evolution of *S. epidermidis* has been observed during the course of prolonged infection, which can impact virulence and antibiotic susceptibility [213]. Through their successful adaptation to harsh conditions, (nosocomial) *S. epidermidis* isolates may have a particularly diverse repertoire of commensal virulence factors readily employed upon host invasion. While there are some markers of virulence more frequently found in infection-associated vs. carriage-associated strains, the pathogenic potential of any given strain cannot be accurately predicted based on genetic analyses. The pathogenicity of *S. epidermidis* as an opportunistic pathogen is strongly influenced by the capacities of the (immunocompromised) host immune response. While preterm immunity has been shown to diverge from that in adults, and even the term infant counterpart, our understanding of the neonatal immune response to *S. epidermidis* infection remains incomplete. Of note, pathogen-specific neonatal (transcriptional) responses have been observed in several studies, indicating that virulence determinants beyond conserved bacterial surface structures modulate host immune responses. Considering that most preterm infants are faced with similar risks and exposure to *S. epidermidis* in the NICU, the host-inherent drivers of *S. epidermidis* sepsis development still need to be determined. What is needed is a comprehensive understanding of which *S. epidermidis* virulence determinants are contributing to invasive neonatal infections in vivo.

Using novel approaches, such as dual RNA-seq [214], host–pathogen interactions between the murine host and the opportunistic pathogens *S. aureus* and *Streptococcus pneumoniae* have recently been characterized [215,216,217,218] and provided new insights into the infection environment. Consequently, it has become evident, that in vitro gene expression studies are not representative of the complex in vivo host–pathogen interplay [217,218]. These studies showed that bacterial virulence gene expression is site-specific, with tissue-tropic virulence phenotypes in vivo [217,218] that are partly driven by the availability of different carbon sources [217]. Host (transcriptional) responses were shown to be isolate-specific and furthermore, host-intrinsic resistance directly related to pathogen gene expression during infection [218].

Blood remains a difficult biological site for analysing host–bacterial interactions during disease [217], given the low bacterial presence, even in culture-positive sepsis. Moreover, neonatal (sepsis) blood samples are particularly limited. However, with advances in the resolution of molecular techniques, such as next generation sequencing platforms, proteomics, and metabolomics, future research may be well suited to tackle these questions and decipher the unique interplay between the preterm neonatal host and the commensal pathogen *S. epidermidis*.

## Figures and Tables

**Figure 1 ijms-23-00860-f001:**
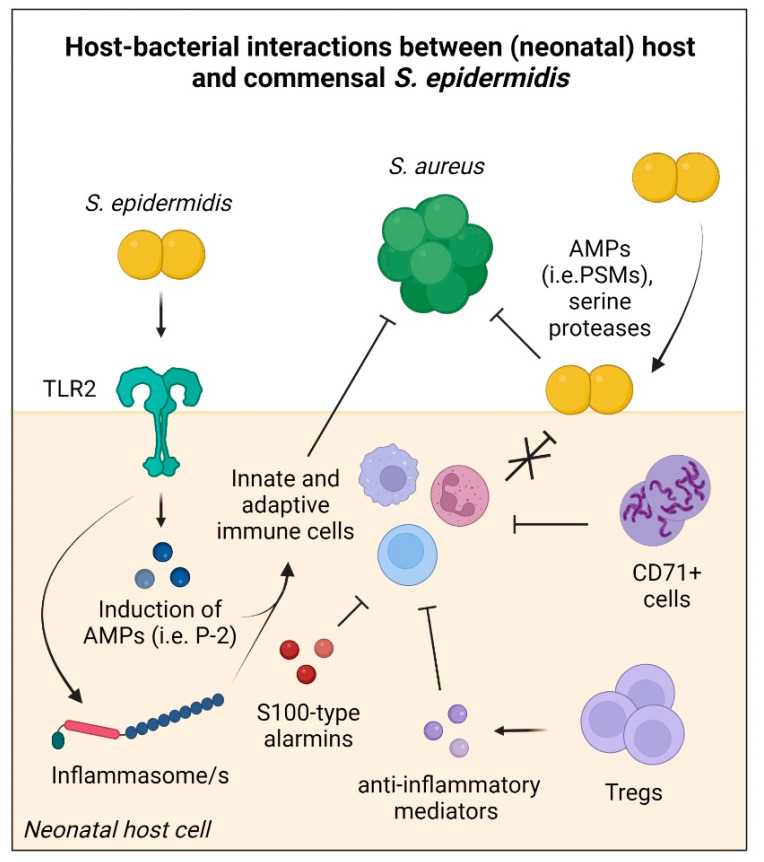
Summary of potential mechanisms underlying the establishment of neonatal tolerance to commensal *S. epidermidis* and its contribution to neonatal host defense. Antimicrobial peptides (AMPs), Phenol-soluble modulins (PSMs), Peforin-2 (P2), Toll-like receptor (TLR), Regulatory T cells (Tregs). Created with BioRender.com.

**Figure 2 ijms-23-00860-f002:**
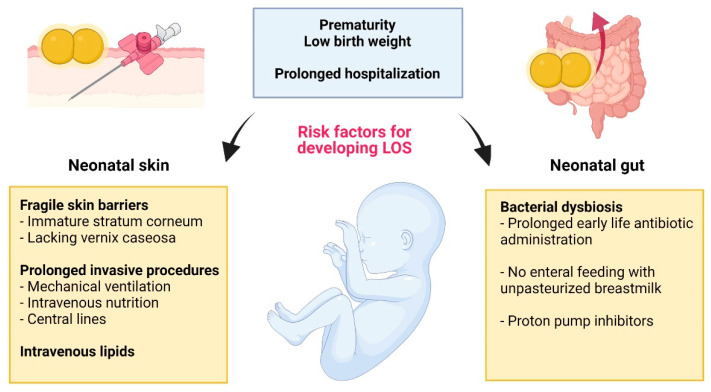
Risk factors associated with skin-derived and gut-derived *S. epidermidis* late-onset sepsis (LOS) in the preterm neonate. Created with BioRender.com.

**Figure 3 ijms-23-00860-f003:**
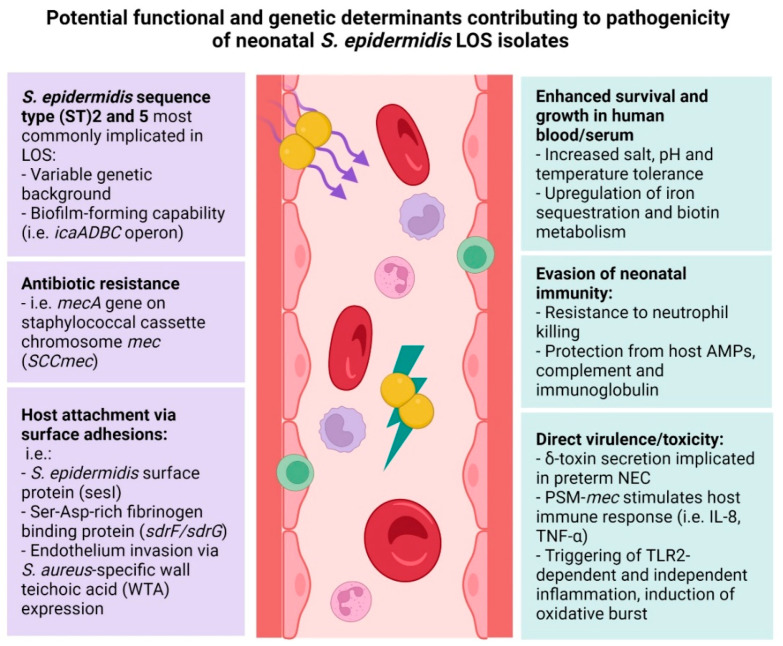
*S. epidermidis* functional and genetic determinants with potential role in late-onset sepsis (LOS pathogenicity). Created with BioRender.com.

## Data Availability

Not applicable.

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
