# Peer review of "Look Who’s Talking: Host and Pathogen Drivers of Staphylococcus epidermidis Virulence in Neonatal Sepsis"

_ijms, 2022, doi:10.3390/ijms23020860_

Round 1

Reviewer 1 Report

Overall this is a very good Review and all my corrections are minor:

Line 57-58: "The impact..." The relevance of this sentence is not explained here and is not further touched until line 572. The authors should consider adding some more information at this point.

Line 76-78: "Currently, the proportions..." It is not clear what the authors mean by proportions, I presume they meant percentages.

Line 150-153: "Other subsets of neonatal immune cells..." Erythrocytes and epithelial cells are not considered immune cells despite their immuno-regulatory roles, I would suggest the authors just alter the beginning of this sentence to "Other subsets of neonatal cells..."

Line 161-162: "In conclusion..." The authors use the word "transition" but fail to mention toward what is the "intrauterine life and tolerance" transitioning. I would suggest completing the sentence or not using that term.

Reviewer 2 Report

In this review, Jouber et al. describe several aspects of the interaction between Staphylococcus epidermidis and the host with a focus on neonatal invasive bacterial infections. The manuscript summarizes an important and impactful topic for a wide audience in the scientific community concerning the clinical emergency of the spread of nosocomial strains of S. epidermidis from a genetic and functional point of view, antibiotic resistance related to infection morbidity and neonatal mortality. Overall, the manuscript is exhaustive and comprehensive. Experimental and clinical results are described in detail, figures are well integrated into the manuscript and the bibliography is adequately included.

Only a few minor points to further improve the publication:

1) If allowed by the journal's policies, a figure summarizing general host-pathogen interactions with more focus on mechanisms of immune modulation would facilitate the reading of sections 2.2 and 2.3 and increase the quality of the manuscript in general.

2) The authors mention the role of circulating proteins, such as the alarmins S100A8/A9, as mediators of the host-pathogen connection. In adults, the serum level of S100A8/A9 is used as one of the most sensitive biomarkers in inflammatory processes, including sepsis. However, S100A8/A9 has been reported not to distinguish septic infants from healthy controls. Along these lines, a paragraph describing circulating markers that are prognostic of infant to adult sepsis should be included in the text (e.g. PTX3; PMID: 31381534 and PMID: 11445697).  
